# Heterogeneous Patterns of Endothelial NF-κB p65 and MAPK c-Jun Activation, Adhesion Molecule Expression, and Leukocyte Recruitment in Lung Microvasculature of Mice with Sepsis

**DOI:** 10.3390/biomedicines12081672

**Published:** 2024-07-26

**Authors:** Zhendong Wang, Erna-Zulaikha Dayang, Peter J. Zwiers, Martha L. Hernandez Garcia, Matthijs Luxen, Matijs van Meurs, Jill Moser, Jan A. A. M. Kamps, Grietje Molema

**Affiliations:** 1Department of Pathology and Medical Biology, Medical Biology Section, University Medical Center Groningen, University of Groningen, Hanzeplein 1, 9713 GZ Groningen, The Netherlands; z.wang02@umcg.nl (Z.W.); ahdezulaikha@unimas.my (E.-Z.D.); ln.marthahgarcia@gmail.com (M.L.H.G.); matthijs.luxen@mpi-muenster.mpg.de (M.L.); m.van.meurs@umcg.nl (M.v.M.); j.moser@umcg.nl (J.M.); j.a.a.m.kamps@umcg.nl (J.A.A.M.K.); 2Department of Critical Care, University Medical Center Groningen, University of Groningen, Hanzeplein 1, 9713 GZ Groningen, The Netherlands

**Keywords:** polymicrobial abdominal sepsis, lung endothelial cells, NF-κB p65, MAPK c-Jun, endothelial adhesion molecules, leukocytes

## Abstract

Background: Sepsis is an uncontrolled systemic inflammatory response to an infection that can result in acute failure of the function of the lung called acute respiratory distress syndrome. Leukocyte recruitment is an important hallmark of acute lung failure in patients with sepsis. Endothelial cells (EC) participate in this process by facilitating tethering, rolling, adhesion, and transmigration of leukocytes via adhesion molecules on their cell surface. In in vivo studies, endothelial nuclear factor kappa-light-chain-enhancer of activated B cells (NF-κB) p65 and mitogen-activated protein kinase (MAPK) c-Jun intracellular signal transduction pathways were reported to regulate the expression of adhesion molecules. Methods: Mice underwent cecal ligation and puncture (CLP) to induce polymicrobial sepsis and were sacrificed at different time points up to 72 h after sepsis onset. Immunohistochemistry and reverse transcription–quantitative polymerase chain reaction (RT-qPCR) analyses were used to determine the kinetics of nuclear localization of p65 and c-Jun in EC, expression and location of adhesion molecules E-selectin and vascular cell adhesion molecule 1 (VCAM-1). Furthermore, the extent and location of leukocyte recruitment were assessed based on Ly6G staining of neutrophils, cluster determinant (CD) 3 staining of T lymphocytes, and CD68 staining of macrophages. Results: In all pulmonary microvascular beds, we identified p65 and c-Jun nuclear accumulation in a subset of endothelial cells within the first 24 h after CLP-sepsis initiation. E-selectin protein was expressed in a subset of microvessels at 4 and 7 h after sepsis initiation, while VCAM-1 was expressed in a scattered pattern in alveolar tissue and microvessels, without discernible changes during sepsis development. CLP-induced sepsis predominantly promoted the accumulation of neutrophils and T lymphocytes 4 and 7 h after disease onset. Neutrophil accumulation occurred in all pulmonary microvascular beds, while T lymphocytes were present in alveolar tissue and postcapillary venules. Taken together, nuclear localization of p65 and c-Jun in EC and neutrophil recruitment could be associated with induced E-selectin expression in the pulmonary microvessels in CLP-septic mice at the early stage of the disease. In alveolar capillaries, on the other hand, activation of these molecular pathways and leukocyte accumulation occurred in the absence of E-selectin or VCAM-1. Conclusions: Endothelial activation and leukocyte recruitment in sepsis-induced lung injury are regulated by multiple, heterogeneously controlled mechanisms, which vary depending on the type of microvascular bed involved.

## 1. Introduction

Sepsis is a dysregulated inflammatory response to infection that can result in multiple organ dysfunction syndrome [1]. The lung is susceptible to infection and can act as a primary source of sepsis or fail as a result of an infection elsewhere in the body, and nearly 21% of patients with sepsis develop acute respiratory distress syndrome (ARDS) [2]. Since the pathophysiology of sepsis-induced acute lung injury is still not fully understood, no specific therapies are available to prevent or treat it. Leukocyte recruitment in the lung is a known hallmark of ARDS [3] and is associated with the activation of microvascular endothelial cells (EC) [4]. Although leukocyte recruitment has been reported to occur in the lungs of septic mice [5], the spatiotemporal pattern of leukocyte recruitment in different pulmonary microvascular beds is unknown.

Lipopolysaccharide (LPS) is a component of the cell wall of gram-negative bacteria and functions as a key mediator of sepsis. In LPS-stimulated EC in vitro, nuclear factor kappa-light-chain-enhancer of activated B cells (NF-κB) and mitogen-activated protein kinase (MAPK) pathways were shown to be activated [6,7]. Activated EC express adhesion molecules such as E-selectin and vascular cell adhesion molecule 1 (VCAM-1), which interact with ligands on leukocytes to facilitate tethering, rolling, adhesion, and transmigration of leukocytes across the endothelium into the tissue [4,8]. 

In EC, LPS and sepsis-related systemically released pro-inflammatory cytokines such as tumor necrosis factor-alpha (TNF-α) activate a complex pattern of kinases [9], which eventually results in the accumulation of transcription factors in the nucleus. Here, they initiate the transcription of DNA sequences that code for, among others, adhesion molecules [10]. In in vitro studies involving LPS or TNF-α, the expression of E-selectin and VCAM-1 in EC was shown to be regulated by NF-κB [11] and MAPK pathways [6,12]. p65 is a transcription factor of the NF-κB pathway, and its inhibition reduces the expression of E-selectin and VCAM-1 in the lung of septic mice [13]. c-Jun is a transcription factor of the activator protein 1 family in the MAPK signaling pathway and is activated in the lung of septic mice [14]. 

Although activation of NF-κB p65 [15] and MAPK c-Jun [14] pathways and expression of adhesion molecules [13] have been reported in the lungs of septic mice, spatial and temporal information regarding them in EC in pulmonary arterioles, alveolar capillaries, and postcapillary venules is currently lacking. In this study, we used a cecal ligation and puncture (CLP) mouse model, which results in polymicrobial sepsis characterized by the release of pro-inflammatory cytokines in the early stages of disease [16]. The kinetics of nuclear localization of p65 and c-Jun proteins in EC was analyzed in pulmonary microvascular beds using immunohistochemical and immunofluorescence double staining. Furthermore, the expression of E-selectin and VCAM-1 at the mRNA level in total lung RNA isolates and at the protein level in the pulmonary microvascular beds was investigated. Lastly, the kinetics of leukocyte recruitment was determined by reverse transcription–quantitative polymerase chain reaction (RT–qPCR), and the spatiotemporal pattern of neutrophil, T lymphocyte, and macrophage accumulation in pulmonary microvascular beds was explored by immunohistochemical staining. This study provides new insights into the spatiotemporal process of recruitment of leukocyte subsets in relation to activation of EC in mouse lungs during the onset and progression of CLP-induced sepsis.

## 2. Materials and Methods

### 2.1. Cecal Ligation and Puncture (CLP) Mouse Model

C57BL/6 male mice (8–12 weeks old, 20–30 g) were purchased from Envigo (Horst, The Netherlands). Lung samples from all mice were obtained from a previous study [17]. Briefly, after anesthetization with isoflurane inhalation, mice were subcutaneously injected with Buprenorphine (0.1 mg/kg, Buprecare, ASTFarma, Oudewater, The Netherlands) to reduce the pain of the mice post-surgery. Subsequently, ligation was performed 1 cm from the top of the mouse cecum using a 5.0 silk suture (5–0 Safil, Braun, Rubí, Spain), and the cecum was punctured once with an 18-gauge needle. Approximately three mm of fecal matter was extruded by gently squeezing the cecum. The cecum was then returned, and the abdominal wound was sutured. Mice in the sham group did not undergo ligation and puncture. After surgery, all mice were immediately s.c. injected with 0.5 mL saline on each abdominal side and then moved to individual 37 °C heated cages for 30 min to recover from surgery, followed by being placed in normal housing conditions. Sham and CLP mice were given access to liquid food. Animals in the groups surviving beyond the 7 h were given antibiotics (Imipinem/Cilastatin, 25 mg/kg, Fresenius Kabi, Bad Homburg vor der Höhe, Germany) and Buprenorphine (0.1 mg/kg) at 10 h after surgery by s.c. injection. Mice were randomly assigned to the control group (untreated; n = 8), sham group (n = 10), and CLP group (n = 13). Mice in sham and CLP groups were sacrificed at 4, 7, 24, or 72 h after surgery. Mouse organs, including the lung, were harvested, snap-frozen on liquid nitrogen, stored at −80 °C, and partly zinc-fixed [18,19], paraffin-embedded (ZnFPE), and stored at room temperature (RT).

### 2.2. Immunohistochemical Staining

Four μm thick lung cryosections were fixed with acetone (100022, Merck Millipore, Darmstadt, Germany) for 10 min and then incubated with H_2_O_2_ (cat. No. #1072090250, Merck Millipore) for 30 min at RT. Next, the sections were incubated with anti-E-selectin and anti-VCAM-1 antibody (Table 1) diluted in 5% (*v*/*v*) fetal calf serum (FCS, cat. No #F6765, Sigma-Aldrich, Saint Louis, MO, USA)/PBS (0.1 mol/L, UMCG Pharmacy, in-house prepared) for 1 h. After washing with PBS, the sections were incubated with rabbit anti-rat antibody (Table 1) diluted in 1% normal mouse serum (NMS, Sanquin, Amsterdam, The Netherlands) in 5% (*v*/*v*) FCS/PBS for 45 min at RT. After washing with PBS, the sections were incubated with the secondary antibody goat anti-rabbit-labeled polymer HRP (cat. No. #K4003, Dako, Carpinteria, CA, USA) for 30 min at RT. After washing with PBS, the sections were incubated with AEC buffer (Table 2) for 9 min, rinsed with demi water, followed by counterstaining with Mayer’s hematoxylin (#109249, 1:4 dilution, Merck Millipore) for 1 min and tap water for washing extra hematoxylin.

Four μm thick paraffin-embedded lungs were deparaffinized with xylene (cat. No. #4055-9005, KLINIPATH, Amsterdam, The Netherlands) for 15 min, followed by rehydration in graded ethanol at 3 min intervals (100%, 96%, and 70% ethanol, Fresenius Kabi, Huis ter Heide, The Netherlands). To retrieve Ly6G antigens (neutrophil surface marker), sections were immersed in 0.1 M Tris-HCl (pH 9.0) (Table 3) and heated in an oven at 70 °C overnight. To retrieve the CD68 (macrophage surface marker) and CD3 antigens (T lymphocyte surface marker), sections were immersed in 1 mM EDTA (pH 8.0) (Table 3) and heated for 15 min in a microwave (600 W, MS-198H, LG, Seoul, Republic of Korea). To retrieve NF-ĸB p65, c-Jun, and ERG epitopes [20,21], the sections were heated in 10 mM Tris/1 mM EDTA (pH 9.0) (Table 3) for 20 min in a microwave at 600 W. After washing with PBS, the sections were incubated with 0.0375% (*v*/*v*) H_2_O_2_ in PBS for 20 min and then incubated with appropriate antibodies (Table 1) in 5% FCS/PBS overnight at 4 °C. After washing with PBS, the Ly6G- and CD3-stained sections were incubated with rabbit anti-rat antibody (Table 1) for 45 min at RT. After washing with PBS, the sections were incubated with the secondary antibody goat anti-rabbit-labeled polymer HRP (cat. No. #K4003, Dako) for 45 min at RT. After washing with PBS, all sections were incubated with AEC (Table 2) and counterstained with Mayer’s hematoxylin, as described above. Isotype controls, including rat IgG1, rat IgG2a, rat IgG2b, and rabbit IgG (Table 1), did not result in any signal.

All sections were mounted with Aquatex (Merck Millipore), and all slides were scanned using a Hamamatsu Nanozoomer 2.0 HT (Hamamatsu Photonics, Hamamatsu, Japan). All mice per group were included in the investigation of E-selectin, VCAM-1, Ly6G, CD3, and CD68. Five mice were randomly selected from each group for staining of p65, c-Jun, and ERG.

### 2.3. Scoring of Immunohistochemical Staining

Scans of the immunohistochemically stained sections from the ZnFPE samples were inspected by eye, with a focus on the pulmonary microvasculature, i.e., those distant from the bronchioles and bronchi. Pulmonary arterioles were discriminated from postcapillary venules based on visible elastic lamina(s), collagen, and smooth muscle cell (SMC) layers in the former, whereas the latter was devoid of a collagen layer and contained a thinner SMC layer than arterioles. Furthermore, arterioles and postcapillary venules could often be identified by the presence of erythrocytes in the lumen. In some cases, sequential sections stained for the endothelial nuclear marker, the ETS-related gene (ERG), were used for microvascular identification. Pulmonary capillaries were identified using immunofluorescence double staining, as described below. As the lung arterioles and postcapillary venules were not easy to distinguish in frozen tissue sections, we called these 2 compartments ‘microvessels’ in the E-selectin and VCAM-1 staining.

To analyze the extent of p65 and c-Jun staining, at least 3~5 arterioles or postcapillary venules were evaluated in each section. To determine staining in alveolar tissue, encompassing alveolar capillaries as well as other cells, at least 10 rectangle areas (240 μm × 390 μm) were randomly selected and evaluated. The scoring values given were absent (no nuclei stained), intermediate (minority of nuclei stained), or extensive (majority of nuclei stained). E-selectin and VCAM-1, as well as Ly6G, CD68, and CD3 staining were also qualitatively scored by visually assessing the scans, with values given being absent, low, or present (cell adhesion molecules)/high infiltration (leukocyte subset). Results per group were summarized.

### 2.4. Immunofluorescence Double Staining

To investigate whether activation of the NF-κB p65 and MAPK c-Jun pathways was present in endothelial cells, sections first underwent deparaffinization and antigen retrieval, as previously described. Subsequently, sections were incubated with 3% (*w*/*v*) bovine serum albumin (cat. No. #A9418, Sigma-Aldrich) in PBS for 30 min and incubated with rabbit anti-mouse p65 or c-Jun antibodies (Table 1) in 5% FCS/PBS overnight at 4 °C in the dark. After washing with PBS, all sections were incubated with Alexa Fluor_555_-donkey anti-rabbit IgG (H+L) (Table 1) in 2% NMS in 5% FCS/PBS for 45 min at RT in the dark. After washing with PBS, the sections were incubated with Alexa Fluor_488_-rabbit anti-ERG (Table 1) in 5% FCS/PBS overnight at 4 °C in the dark. After washing with PBS, all sections were incubated with TrueVIEW Autofluorescence Quenching Kit (cat. No. #SP-8400-15, Vector Laboratories, Newark, CA, USA) for 3 min at RT in the dark. After washing with PBS, all sections were incubated with DAPI in 5% FCS/PBS for 10 min at RT in the dark. VECTASHIELD^®^ Vibrance™ Antifade Mounting Medium (cat. No. #SP-8400-15, Vector Laboratories) was used to mount all the sections. All images were taken with equal exposure times using a Leica DM4000B fluorescence microscope equipped with a Leica DFC345FX digital camera (Leica Microsystems Ltd., Wetzlar, Germany) and Leica LAS V4.5 Image Software at 200× magnification. To ensure signal specificity, sections were incubated with anti-p65/Alexa Fluor_555_-donkey anti-rabbit Abs, anti-c-Jun/Alexa Fluor_555_-donkey anti-rabbit Abs, or Alexa Fluor_488_-rabbit anti-ERG, followed by DAPI staining. Analysis in all three channels always showed 2 colors, DAPI blue and either target red or ERG green.

### 2.5. Gene Expression Analysis by RT-qPCR 

Twenty-five lung cryosections (10 μm per section) were collected in 1.5 mL RNase-free tubes. Total RNA isolation, RNA concentration detection, and RNA integrity analysis were performed as described previously [17]. RNA was reverse-transcribed to cDNA using SuperScript^®^ III reverse transcriptase (Invitrogen, Waltham, MA, USA) and random hexamer primers (Promega, Leiden, The Netherlands). Subsequently, cDNA was used for qPCR using the ViiA 7 real-time PCR System (Applied Biosystems, Nieuwerkerk aan den IJssel, The Netherlands). The Assay-on-Demand FAM-MGB-labeled probes (TaqMan gene expression, Thermo Fisher Scientific, Waltham, MA, USA) used in the qPCR analysis are listed in Table 4. QuantStudio Real-Time PCR software (v. 1.3, Applied Biosystems) was used to analyze data to obtain average quantification cycle (Cq) values. Expressed genes were normalized to the reference gene *Gapdh*, resulting in the ∆Cq value. The relative mRNA level was calculated using 2^−∆Cq^.

### 2.6. Myeloperoxidase (MPO) ELISA

Ten μm thick lung cryosections per mouse were lysed using RIPA lysis buffer (50 mM Tris pH 8.0, 100 mM NaCl (cat. No. #648311 and #106404, both from Merck Millipore), 0.1% (*w*/*v*) sodium dodecyl sulphate, 0.5% (*v*/*v*) sodium deoxycholate, and 1% (*v*/*v*) IGEPAL^®^ (cat. No. #05030, #30970, and #542334, all from Sigma-Aldrich)) containing phosphatase and protease inhibitors (cat. No. #04906845001 and #04693124001, both from Roche, Almere, The Netherlands). A protein assay kit (cat. No. #500-0115, Bio-Rad, Hercules, CA, USA) was used for protein quantification according to the manufacturer’s instructions. Subsequently, an MPO ELISA kit (cat. No. #HK210-02, Hycult Biotechnology, Uden, The Netherlands) was used to detect myeloperoxidase (MPO) concentrations in the lysates according to the manufacturer’s instructions.

### 2.7. Statistical Analyses

One-way analysis of variance (ANOVA) with Bonferroni post-hoc comparison was used for data analysis. GraphPad Prism v.10.0 software (GraphPad Software, Solana Beach, CA, USA) was used for all statistical analyses. Differences were considered statistically significant when *p* < 0.05.

## 3. Results

### 3.1. NF-κB p65 Was Activated in EC in All Pulmonary Microvascular Beds during the First 24 h of Sepsis

Activation of the p65 protein has been reported to occur in the lung of mice with CLP-sepsis [22], yet the spatiotemporal pattern of nuclear p65 in the pulmonary microvascular arterioles, alveolar capillaries, and postcapillary venules is unknown (Figure 1A). 

Immunohistochemical staining showed that in control conditions, the p65 protein was extensively present in the cytoplasm in most pulmonary cells, with occasional staining of nuclei in arteriolar EC, in cells in alveolar tissue, and in postcapillary venular EC (Figure 2A). While an increase in p65-positive nuclei was observed within the first 24 h after sepsis initiation, the predominant CLP-induced change was an increase in positive nuclei in alveolar tissue and postcapillary venules at 4 h (Figure 2A). In the first 24 h after sham operation, the extent of p65-positive nuclei was comparable to that in the control group (Appendix A). At 72 h after CLP-induced sepsis initiation, p65 nuclear localization was comparable to that in the sham group (Appendix A).

To investigate whether p65 nuclear staining in alveolar tissue could be attributed to EC, we performed immunofluorescence double staining for p65 and ERG, an endothelial nuclear marker (Figure 1B). This revealed that p65 accumulated in the nuclei of a subset of EC in the capillaries in the first 24 h after CLP-induced sepsis initiation, in addition to p65 accumulation in the nuclei of non-endothelial cells (Figure 2B). 

Thus, NF-κB p65 pathway activation occurred in a subset of EC in all pulmonary microvascular compartments during the first 24 h after CLP-induced sepsis initiation.

### 3.2. MAPK c-Jun Was Activated in EC in All Pulmonary Microvascular Beds until 24 h after Sepsis Onset

Western blot data in a previous report showed that c-Jun protein was activated in mouse lungs 6 h after the onset of sepsis [14], yet neither kinetics nor location of the activated MAPK c-Jun were reported. Immunohistochemical analysis showed that in control conditions, sparse c-Jun-positive nuclei were detected in all pulmonary microvascular compartments (Figure 3A).

Following CLP-induced sepsis, more nuclei became c-Jun positive. In alveolar tissue and postcapillary venules, this increase was only observed at 4 h post-CLP, while scarce positive nuclei were present in the sham group (Figure 3A and Appendix A). Twenty-four hours after CLP-induced sepsis initiation, a discernible difference in c-Jun nuclear localization in alveolar tissue became apparent in the majority of mice, compared to the sham groups. At 72 h after CLP-induced sepsis started, nuclear c-Jun in the lung returned to basal levels comparable to those in the control and sham groups (Appendix A). Double immunofluorescence staining for c-Jun and ERG showed that in the first 24 h of CLP-induced sepsis, positive nuclei in alveolar capillaries were partly derived from EC and partly from non-endothelial cells (Figure 3B). 

In summary, CLP-induced sepsis resulted in MAPK c-Jun pathway activation in a subset of EC in all pulmonary microvascular beds in the first 24 h after disease initiation. 

### 3.3. In the Early Stage of CLP-Sepsis, E-Selectin Was Induced in Pulmonary Microvessels, Whereas VCAM-1 Expression Was Hardly Affected

Endothelial NF-κB p65 and MAPK c-Jun pathways were reported to be associated with induced expression of adhesion molecules [12,13,25]. We thus hypothesized that the spatiotemporal expression of E-selectin and VCAM-1 proteins in pulmonary microvascular beds in sepsis would coincide with p65 and c-Jun pathway activation. *Sele* mRNA levels were increased in the lung 4 and 7 h after the onset of CLP-induced sepsis and thereafter declined to levels comparable to those in control and sham mice (Figure 4A). Interestingly, E-selectin protein was present mainly in some pulmonary microvessels, but not in alveolar tissue, at 4 and 7 h after CLP-induced sepsis initiation (Figure 4B). No differences in E-selectin expression levels were detected at later time points (Figure 4B). In sham groups, E-selectin expression was absent at all time points (Figure 4B).

The mRNA levels of *Vcam1* showed a minor increase at 4 h after CLP-induced sepsis started, compared to sham mice. Thereafter, in septic mice, it decreased to levels similar to those observed in sham mice. Notably, at 72 h post-CLP, *Vcam1* mRNA levels were decreased compared to sham mice (Figure 5A). In control, sham, and CLP-septic mice, VCAM-1 protein was mainly expressed in pulmonary microvessels, with a scattered pattern in alveolar tissue. No changes were observed in response to sepsis (Figure 5B). These data demonstrated that in the early stage of CLP-induced sepsis, expression of E-selectin protein was most pronouncedly induced in a subset of microvessels composed of arterioles and postcapillary venules, while VCAM-1 protein was constitutively expressed in these microvessels and no changes were observed during sepsis onset and progression. 

### 3.4. Pulmonary Leukocyte Recruitment Occurred in a Time-Dependent Manner during CLP-Induced Sepsis

We next analyzed leukocyte recruitment into the lung by assessing the mRNA levels of the pan-leukocyte marker *Cd45* [26], which increased at 4 and 7 h after CLP-sepsis initiation compared to sham (Figure 6A). Levels of myeloperoxidase protein (MPO), a heme-containing peroxidase expressed mainly by neutrophils to participate in pathogen clearance [27], were also increased at 4 and 7 h after CLP-induced sepsis initiation, and decreased thereafter (Figure 6B). Thus, sepsis resulted in the recruitment of leukocytes, including neutrophils, into the lung at early time points after CLP surgery, corroborating previous studies [28].

### 3.5. The Recruitment of Leukocyte Subsets Showed Different Spatiotemporal Patterns in Mouse Pulmonary Microvascular Beds during CLP-Induced Sepsis

We next investigated if and where neutrophils were recruited during sepsis in mouse pulmonary microvascular beds. While immunohistochemical staining of Ly6G indicated that under control conditions only sparse neutrophils were present in alveolar tissue and postcapillary venules (Figure 7), CLP-induced sepsis resulted in an increase in neutrophil accumulation in all microvascular compartments at 4 h (Figure 7). A decrease at 7 and 24 h in alveolar tissue and postcapillary venules to levels comparable to those seen in sham mice was followed by a second wave of increased accumulation in alveolar tissue at 72 h (Figure 7). In all sham groups, sparse neutrophils were present in alveolar tissue and postcapillary venules (Figure 7).

As previous studies reported that murine sepsis led to T lymphocyte [29] and macrophage [30] recruitment to the lung, we studied the kinetics and location of these leukocyte subsets as well. At the mRNA level, *Cd3e* levels representing T lymphocytes did not statistically differ between CLP-sepsis and sham groups, although at the latest time points after sepsis initiation, in the sham groups an increased influx of T lymphocytes was observed compared to CLP-septic mice (Figure 8A). Immunohistochemical staining of CD3 indicated that T lymphocytes were present in alveolar tissue and postcapillary venules in control conditions, with an increase in accumulation in alveolar tissue mainly at 4 and 7 h (Figure 8B). Accumulation of T lymphocytes in alveolar tissue showed an increase again at 72 h in CLP-septic mice compared to 24 h after sepsis initiation (Figure 8B). In all sham groups, sparse T lymphocytes were present in alveolar tissue (Figure 8). T lymphocyte accumulation in postcapillary venules was minor and did not show a discernible difference between control, sham, and CLP-septic mice (Figure 8B). *Cd68* mRNA levels in the lung did not change during the onset and progression of sepsis (Appendix A), nor was it affected by sham surgery, which was corroborated by immunohistochemistry (Appendix A). Thus, CLP-induced sepsis resulted in the accumulation of neutrophils in all pulmonary microvascular beds and T lymphocytes in alveolar tissue at the early stage of the disease, whereas no changes in macrophage accumulation were observed throughout the study period.

## 4. Discussion

Leukocyte recruitment is a hallmark of sepsis-ARDS [3] and is mediated by adhesion molecules that are expressed by EC following the activation of dedicated signaling pathways [13]. Previous studies have revealed that the endothelial NF-κB p65 and MAPK c-Jun pathways are engaged in inducing E-selectin and VCAM-1 expression [12,13,31]. The current study aimed to investigate the location of activated p65 and c-Jun pathways in EC, of expression of E-selectin and VCAM-1 as representatives of endothelially restricted adhesion molecules, and of leukocyte recruitment in mouse pulmonary microvascular beds in CLP-induced sepsis in time. We showed that nuclear localization of p65 and c-Jun was present in EC in all pulmonary microvascular beds in the first 24 h after sepsis initiation, albeit not in all EC in these microvascular beds. At 4 and 7 h following sepsis, E-selectin was expressed in a subset of the microvessels yet was absent from alveolar capillaries. VCAM-1 protein, on the other hand, was constitutively expressed mainly in microvessels and in a scattered manner in alveolar capillaries, which did not change during sepsis development. In addition to this heterogeneous microvascular response, heterogeneity in leukocyte subset accumulation was also observed. Neutrophils accumulated in all pulmonary microvascular beds mainly at 4 h, and T lymphocytes accumulated predominantly in alveolar tissue at 4 and 7 h after the onset of the disease, while macrophage content of the lung did not change. In summary, in mouse lungs, there is a high level of microvascular heterogeneity in response to sepsis in the CLP model, with no clear relationship between activation of the endothelial NF-κB p65 and MAPK c-Jun pathways and inflammation-associated adhesion molecule expression, nor with leukocyte recruitment.

Endothelial NF-κB p65 and MAPK c-Jun pathways have been shown to regulate E-selectin and VCAM-1 expression [12,13]. Although the current study only assessed the nuclear translocation of proteins involved in this transcriptional control and not transcription factor activity per se, it is expected that the translocated p65 and c-Jun proteins have functional implications. Within this premise, it was unexpected that the nuclear localization of p65 and c-Jun was unaccompanied by the expression of E-selectin and VCAM-1 in alveolar capillaries. It is possible that, in vivo, engagement of endothelial NF-κB p65 is also related to permeability control in the lung, as has been reported in CLP-septic mice [13]. Investigating permeability changes was not the aim of our current study; however, it would be worthwhile in the future to further explore the role of endothelial NF-κB p65 in the expression of molecules associated with endothelial junction control both at mRNA and protein levels, such as vascular endothelial cadherin, platelet endothelial cell adhesion molecules, and Tie2 [32]. Investigating the role of other transcription factors involved in endothelial inflammatory activation and vascular permeability relies on available antibodies for tissue applications. Combining this with laser microdissection or single-cell isolation, proteomics, phospho-proteomics, or transcription factor-focused digital spatial profiling will be crucial to reveal the full molecular status of the response machinery of EC and to identify potential targets for therapeutic intervention [33]. In addition, we attempted to quantify nuclear p65 and c-Jun proteins, as well as leukocyte subsets. As neither hand-counting nor software programs (Image J, TissueFAXS Imaging Software, Aperio ImageScope, using 2023 versions) could accurately recognize each positive nucleus, respectively, cell, and for leukocyte subset semi-quantitation by flow cytometry no fresh tissue was available, a descriptive method was taken to report these parameters.

A heterogeneous pattern of nuclear accumulation of p65 and c-Jun was observed in EC in the lung microvessels of mice with CLP-induced sepsis. Whether the activation of these signal transduction pathways is associated with the induced expression of E-selectin is currently unknown. Immunofluorescence double staining of E-selectin and p65 or c-Jun could provide valuable information in this respect. However, antibodies against mouse E-selectin were not suitable for staining zinc-fixed and paraffin-embedded tissues. Since frozen tissue sections were histologically and morphologically inferior for identifying EC nuclei, we are currently unable to shed light on this issue. What remains unequivocally revealed in this study is that there is a significant level of heterogeneity in EC inflammatory responses to sepsis within one microvascular segment and between microvascular segments. Expression of VCAM-1 protein in a subset of microvascular segments in a quiescent state remains enigmatic, both functionally and in relation to the molecular mechanisms underlying expression control. A variation between mice in nuclear accumulation of c-Jun in the lung microvascular beds was also observed, 24 h after CLP-induced sepsis started, with c-Jun-positive nuclei showing a discernable increase in three out of five mice, which supports a prior report by Lewis et al. [34].

Post-transcriptional control could be an explanation for the observation that induced expression of E-selectin was not accompanied by nuclear localization of endothelial NF-κB p65 and MAPK c-Jun. Mature microRNAs (miRs) are short noncoding RNAs that bind to (partially) complementary sequences of target mRNAs, leading to the inhibition of protein synthesis by degradation or translation repression [35]. A role of miR-181b in E-selectin expression has been reported in EC in vitro [36] and in the lung of LPS-treated mice [37]. Future studies assessing the levels of miR-181b in each pulmonary microvascular bed via laser microdissection and RT-qPCR could provide valuable information on whether miRs could theoretically explain the expression pattern of E-selectin protein [35].

Recruitment of leukocytes, including neutrophils, was previously reported to occur early in the lung of mice with CLP-sepsis [38]. Our immunohistochemical data also revealed increased T lymphocyte accumulation in alveolar tissue in the first 7 h after the onset of CLP-induced sepsis. Neutrophil adhesion to LPS-stimulated endothelial cells in vitro is associated with the expression of E-selectin and VCAM-1 [39,40], and a role for VCAM-1 in T lymphocyte adhesion in vitro has been reported in TNF-α-stimulated endothelial cells [41]. These adhesion molecules bind to ligands such as P-selectin glycoprotein ligand-1 and very late antigen-4 on neutrophils/T lymphocytes to facilitate their recruitment [4,8]. However, neutrophil recruitment was previously shown to be independent of E-selectin expression in alveolar capillaries of mice treated with *Streptococcus pneumoniae* [42]. Instead, the accumulation of neutrophils and T lymphocytes in alveolar capillaries may be related to mechanical trapping [43], as the diameter of alveolar capillaries is smaller than that of these leukocyte subsets [23,24]. In the microvessels, on the other hand, increased E-selectin expression was accompanied by the recruitment of neutrophils, but not T lymphocytes, at the early stage after CLP-induced sepsis initiation. Investigating the relationship between adhesion molecules and leukocytes may be important to clarify the molecular processes underlying leukocyte recruitment and associated organ damage. This could be further explored by the application of E-selectin inhibitors, such as GMI-1271 [44] or Sele knockout mice. In the latter case, the extent and location of knockdown should be carefully investigated, as we and others have reported heterogeneous inducible knockout or target genes in such mouse models [13,45]. In pulmonary microvessels of mice with CLP-sepsis, one reason why VCAM-1 protein expression is likely not involved in neutrophil and T lymphocyte accumulation could be related to the endothelial surface layer (ESL) composed of the glycocalyx. ESL is thought to prevent neutrophils from binding to VCAM-1 on endothelial cells. In LPS-induced endotoxemia, ESL was incompletely lost in the lung, thereby possibly still masking adhesion molecule sites on the outer surface of the endothelial cell membrane [46]. In our study, the ESL in the microvessels could also be incompletely lost, thereby blocking VCAM-1 binding to very late antigen 4 on neutrophils/T lymphocytes in mice with CLP-induced sepsis. The kinetics of changes in ESL thickness in the microvessels in CLP-septic mice using simultaneous brightfield and fluorescent microscopy could be investigated in the future [46].

## 5. Conclusions

Our study showed that in alveolar tissue, CLP-induced sepsis resulted in the activation of the endothelial NF-κB p65 and MAPK c-Jun pathways. This was not accompanied by E-selectin and VCAM-1 protein expression, yet a simultaneous increase in neutrophil and T lymphocyte accumulation occurred in alveolar tissue in the early stages after the disease started. In arterioles and postcapillary venules, where activation of endothelial NF-κB p65 and MAPK c-Jun pathways and neutrophil recruitment took place in the early stage of the disease, E-selectin was expressed. VCAM-1 protein expression in pulmonary microvascular beds did not change in this sepsis model. The complexity of the molecular responses of endothelial cells in their in vivo pulmonary microvascular niche and their role in neutrophil and T lymphocyte recruitment requires further investigation that focuses on the heterogeneous molecular and spatiotemporal nature of the response.

## Figures and Tables

**Figure 1 biomedicines-12-01672-f001:**
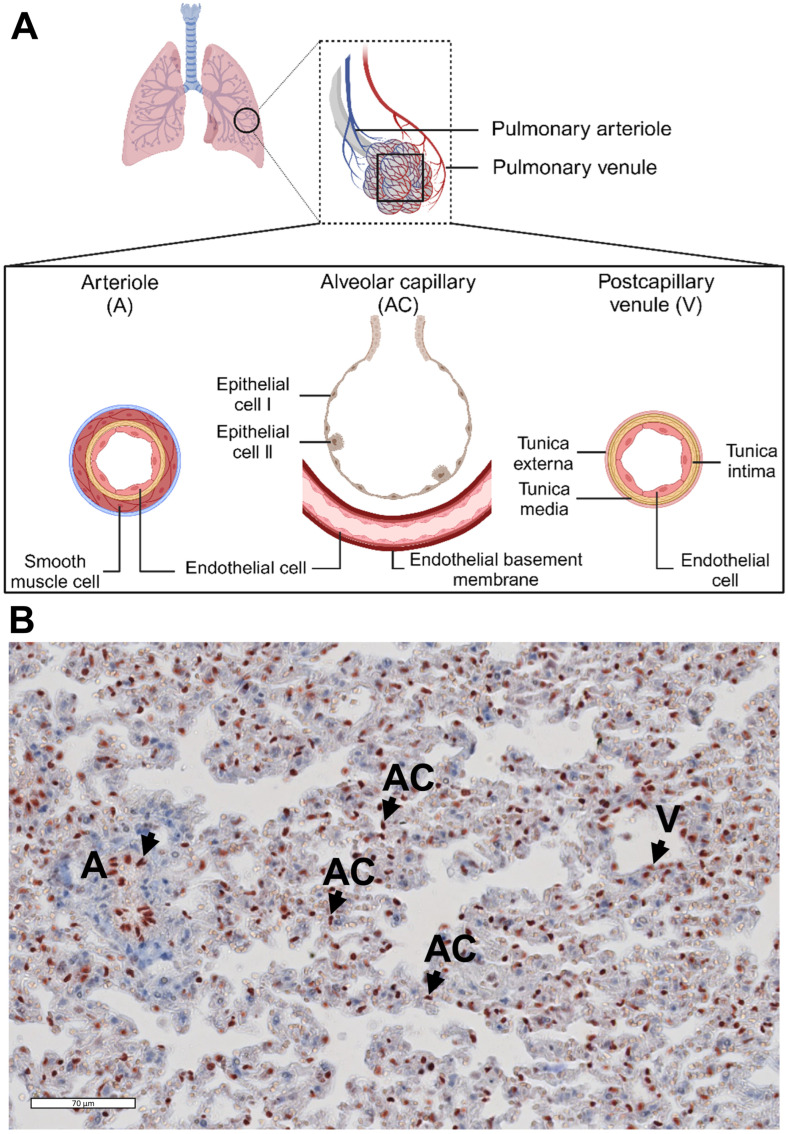
Schematic overview of the structure of pulmonary microvascular beds and location of the endothelium. (**A**) The pulmonary microvascular beds in mice include arterioles (A, diameter: 22.7 ± 7.8 µm), which deliver oxygen-low blood to the extensive network of alveolar capillaries [23]. Alveolar capillaries (AC, diameter: 6 ± 2 µm) exist between the walls of the adjacent alveoli and are a place for gas exchange [24]. Postcapillary venules (V, diameter: 26.4 ± 13.1 µm) are small veins after the alveoli that carry oxygen-rich blood to the left atrium [23]. As the lung arterioles and postcapillary venules are not easy to distinguish in frozen tissue, we called these 2 compartments ‘microvessels’ in relation to assessment of E-selectin and VCAM-1 staining. Bronchial vessels are not included in this study. (**B**) Immunohistochemical staining of the endothelial nuclear marker ETS-related gene (ERG) [21] showed where endothelial cells are located in pulmonary microvascular beds in mice. Arrows indicate ERG-positive endothelial nuclei.

**Figure 2 biomedicines-12-01672-f002:**
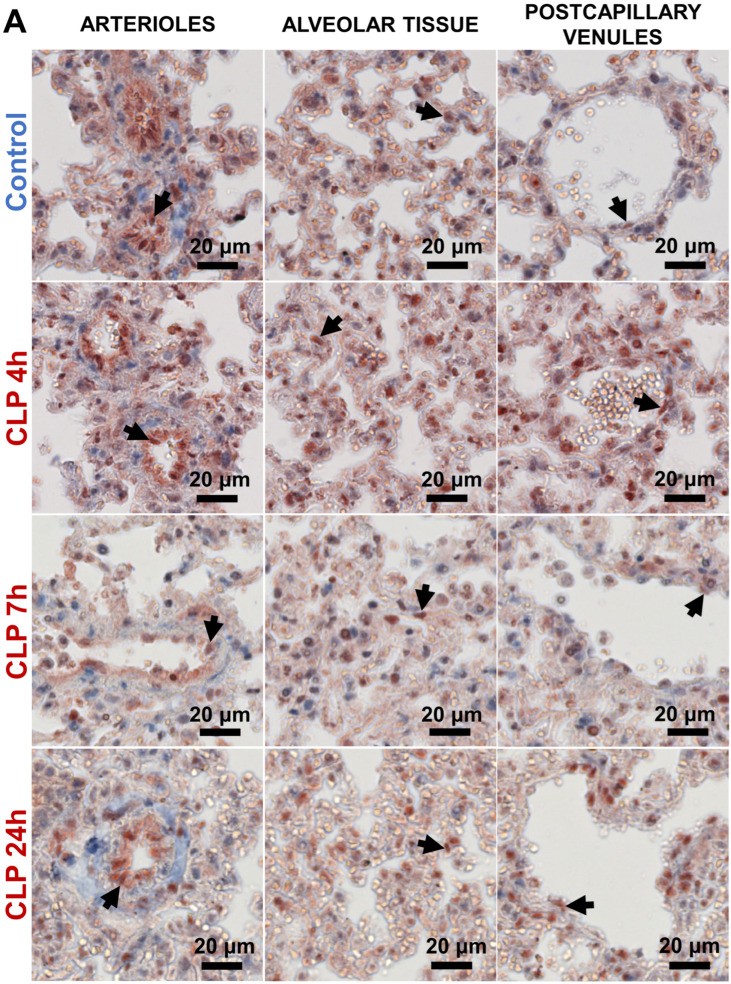
Nuclear localization of p65 protein in the pulmonary microvasculature of mice in the first 24 h of CLP-induced sepsis. (**A**) Immunohistochemical staining of p65 showed nuclear localization in arterioles (**A**), alveolar tissue (AT), and postcapillary venules (V) of the lung in control and CLP-septic mice. Black arrows indicate the red p65 positive nuclei. Images are representative of five mice per group. (**B**) Immunofluorescence double staining of p65 (red) and ERG (green), combined with DAPI (blue, nuclear staining) in alveolar capillaries of mice 4, 7, and 24 h after CLP-induced sepsis initiation. White arrows (blue + red + green) indicate nuclear colocalization of endothelial ERG and p65, turquoise arrows (blue + green) indicate ERG-positive endothelial nuclei, and purple arrows (blue + red) indicate p65-positive non-endothelial nuclei. Images are representative of three mice per group.

**Figure 3 biomedicines-12-01672-f003:**
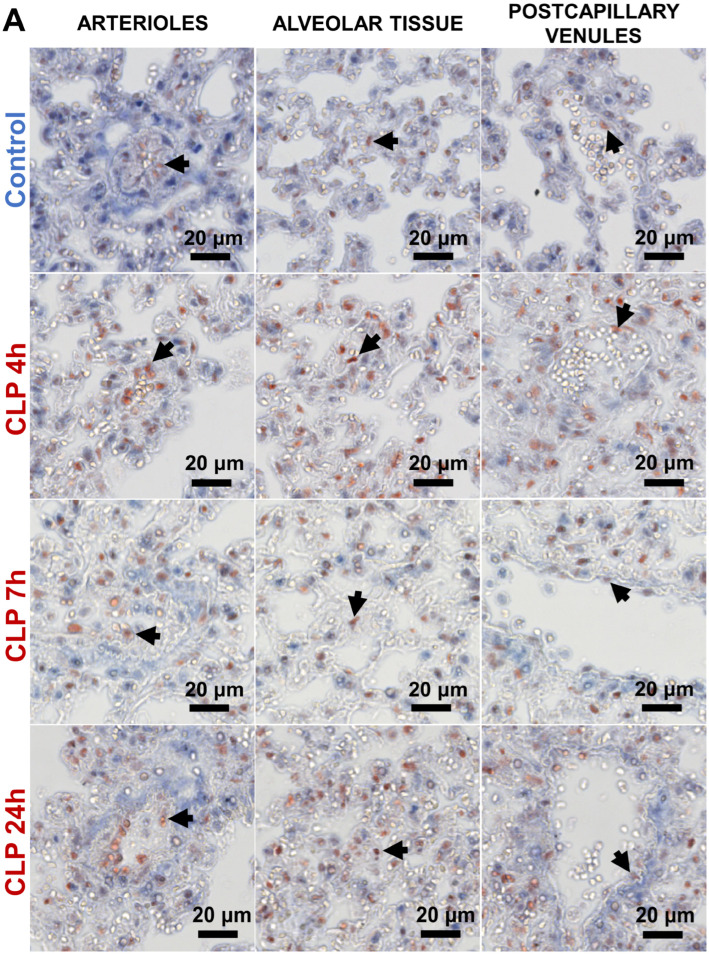
Nuclear localization of c-Jun protein in the pulmonary microvasculature of mice in the first 24 h of CLP-induced sepsis. (**A**) Immunohistochemical staining of c-Jun showed nuclear localization in arterioles (A), alveolar tissue (AT), and postcapillary venules (V) of the lung in control and CLP-septic mice. Black arrows point at the red c-Jun positive nuclei. Images are representative of five mice per group. (**B**) Immunofluorescence double staining of c-Jun (red) and ERG (green), combined with DAPI (blue, nuclear staining) in alveolar capillaries of mice 4, 7, and 24 h after CLP-induced sepsis. White arrows (blue + red + green) indicate nuclear colocalization of endothelial ERG and c-Jun, turquoise arrows (blue + green) indicate ERG-positive endothelial nuclei, and purple arrows (blue + red) indicate c-Jun-positive non-endothelial nuclei. Images are representative of three mice per group.

**Figure 4 biomedicines-12-01672-f004:**
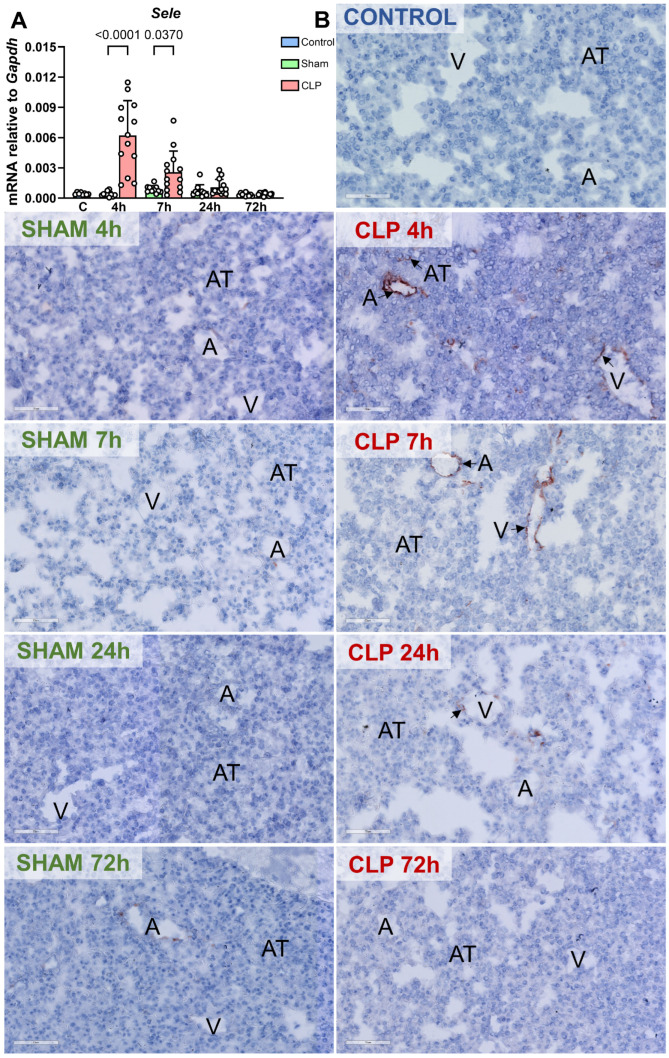
Kinetics of *Sele* mRNA expression and location of E-selectin protein in the lung of mice with CLP-induced sepsis. (**A**) Kinetics of *Sele* in the lung of control (C, blue, n = 8), sham (green, n = 10), and CLP-septic (red, n = 13) mice was assessed by RT-qPCR. The graph shows individual values and mean ± SD, *p* < 0.05. (**B**) Immunohistochemical staining of E-selectin protein, revealing the location in the lung of control, sham, and CLP-septic mice at different time points. Black arrows point at the E-selectin protein. Arteriole- and venule-like microvascular beds are called ‘microvessels’, and alveolar tissue is named ‘AT’. Images are representative of 8~13 mice per group.

**Figure 5 biomedicines-12-01672-f005:**
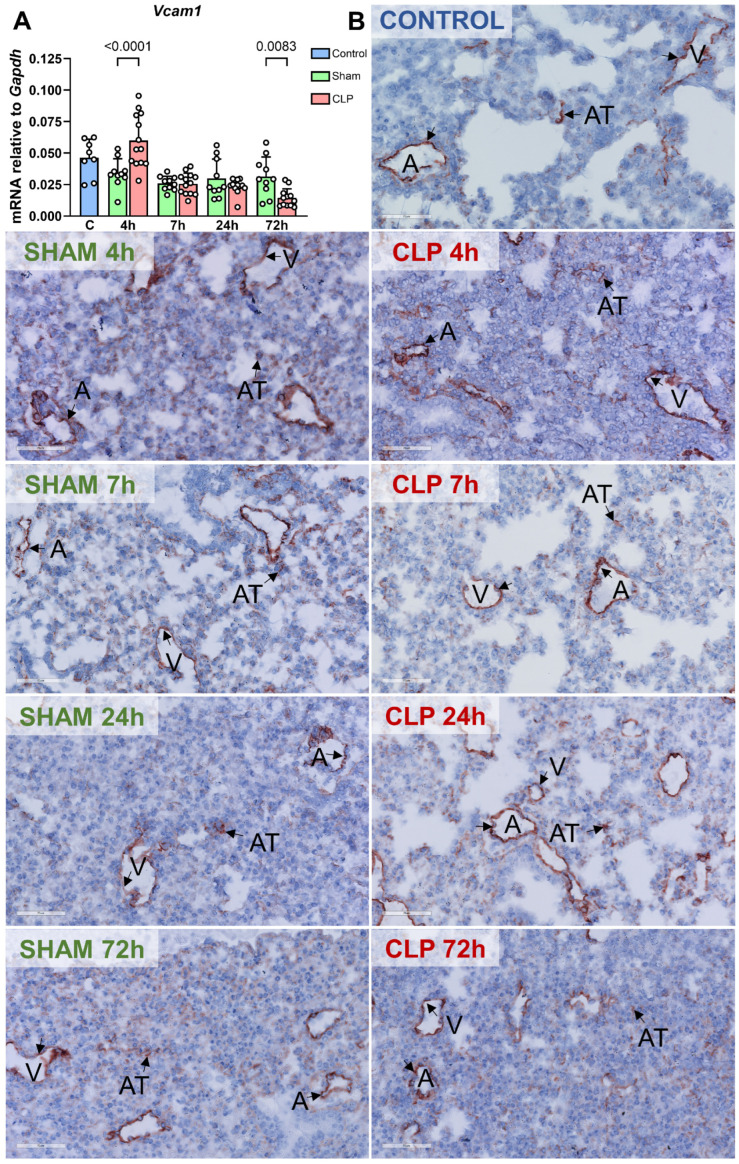
Kinetics of *Vcam1* mRNA expression and location of VCAM-1 protein in the lung of mice with CLP-sepsis. (**A**) Kinetics of *Vcam1* in the lung of control (C, blue, n = 8), sham (green, n = 10), and CLP-septic mice (red, n = 13) was assessed by RT-qPCR. Graph shows individual values and mean ± SD, *p* < 0.05. (**B**) Immunohistochemical staining of VCAM-1 protein, revealing its location in the lungs of control, sham, and CLP-septic mice at different time points. Black arrows point at VCAM-1 protein. All arteriole- and venule-like microvascular beds are called ‘microvessels’, and alveolar tissue is named ‘AT’. Images are representative of 8~13 mice per group.

**Figure 6 biomedicines-12-01672-f006:**
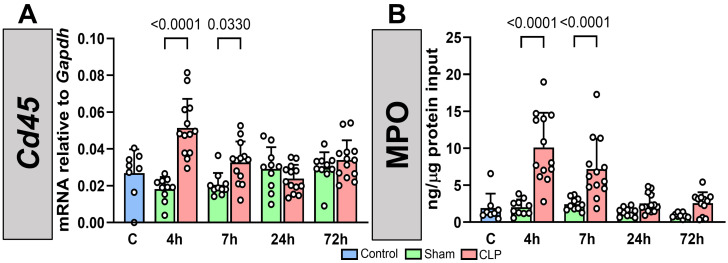
CLP-induced sepsis resulted in leukocyte accumulation in the lung at early time points. (**A**) Expression kinetics of pan-leukocyte molecule *Cd45* in the lung of control (C, blue, n = 8), sham (green, n = 10), and CLP-septic (red, n = 13) mice was determined by RT-qPCR. (**B**) Kinetics of myeloperoxidase (MPO) protein content in the lung as assessed by ELISA. Graphs show individual values and mean ± SD.

**Figure 7 biomedicines-12-01672-f007:**
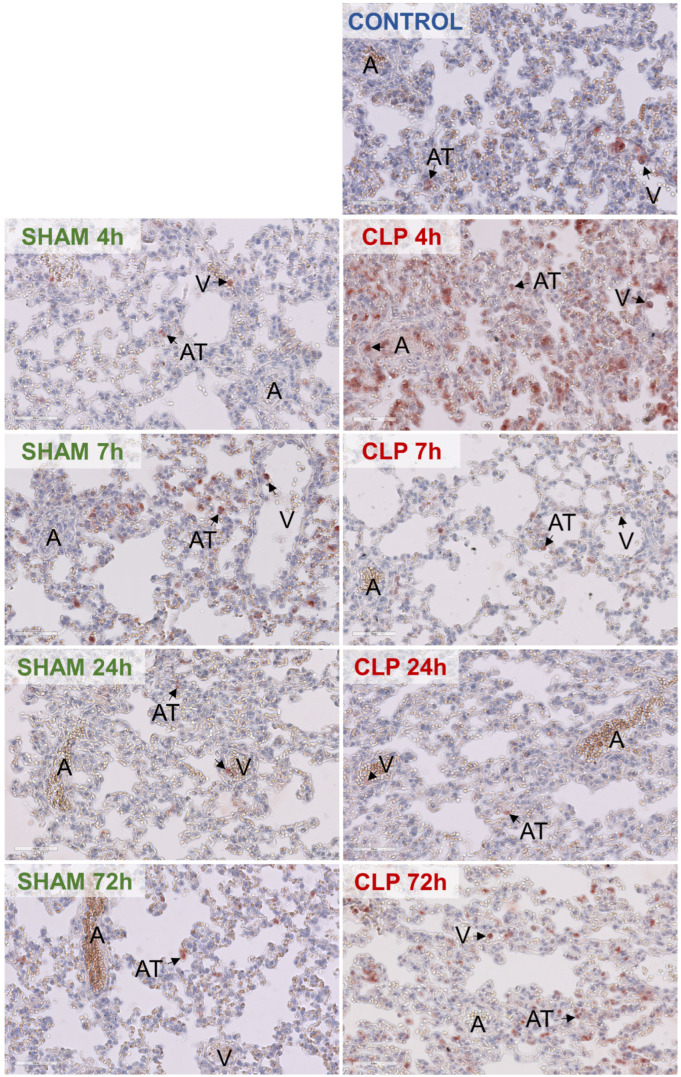
Distribution of neutrophils in pulmonary microvascular beds of mice during CLP-induced sepsis initiation and progression. Immunohistochemical staining of Ly6G was performed to detect their location in the lung of control, sham, and CLP-septic mice at indicated time points. Black arrows point at neutrophils. Arterioles: ‘A’, alveolar tissue ‘AT’, and postcapillary venules: ‘V’. Images are representative of 8~13 mice per group.

**Figure 8 biomedicines-12-01672-f008:**
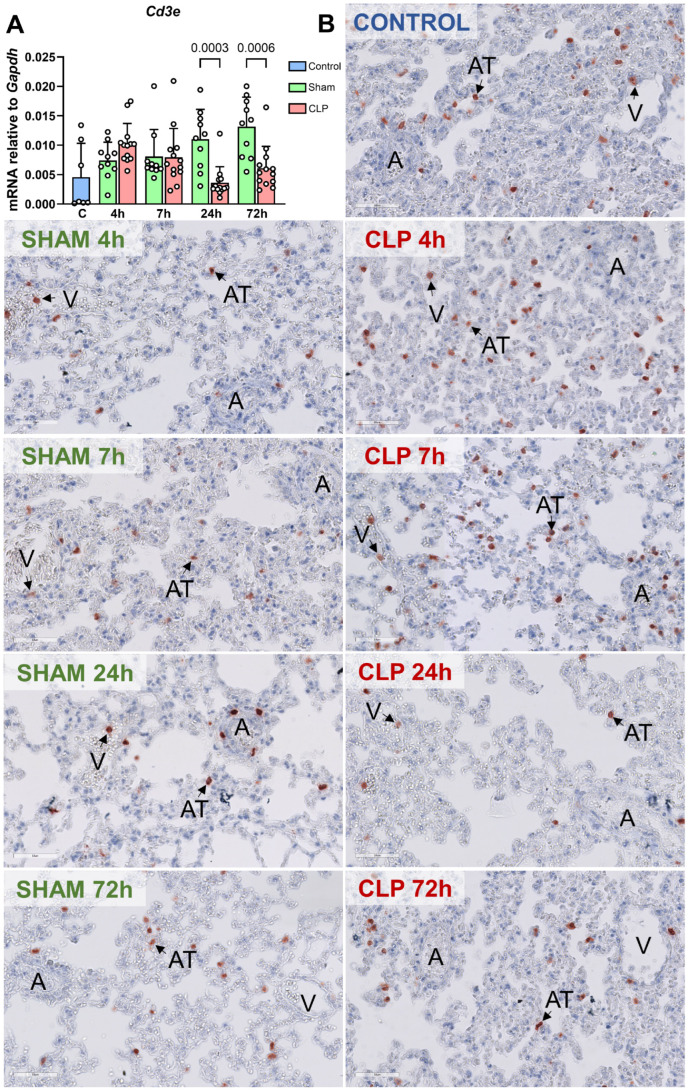
Distribution of T lymphocytes in pulmonary microvascular beds of mice during CLP-induced sepsis initiation and progression. (**A**) The kinetics of T lymphocyte molecule *Cd3e* in the lung of control (C, blue, n = 8), sham (green, n = 10), and CLP-septic (red, n = 13) mice was detected by RT-qPCR. Graphs showed individual values and mean ± SD, *p* < 0.05. (**B**) Immunohistochemical staining of CD3-positive T lymphocytes was performed to determine their location in the lungs of control, sham, and CLP-septic mice at indicated time points. Black arrows point at T lymphocytes. Arterioles: ‘A’, alveolar tissue ‘AT’, and postcapillary venules: ‘V’. Images are representative of 8~13 mice per group.

**Table 1 biomedicines-12-01672-t001:** Antibodies used for immunohistochemical staining or immunofluorescence double staining.

Epitopes and Antibodies	Species or Fluorophore	Working Concentration	Cat. No	Company
		(μg/mL)		
E-selectin	Rat anti-mouse	10	clone MES-1	Kind gift from Dr. Derek Brown
VCAM-1		10	CBL1300	Merck Millipore
Ly6G		3.33	NBP2-00441	Novus Biologicals, Centennial, CO, USA
CD3		0.83	MCA1477	Bio-Rad
IgG (H+L)	Rabbit anti-rat	3.33	AI-4001	Vector Laboratories
CD68	Rabbit anti-mouse	2.5	ab125212	Abcam, Waltham, MA, USA
p65		0.53	8242	Cell signaling, Danvers, MA, USA
c-Jun		0.48	9165	
ERG		4.4	ab92513	Abcam
ERG	Alexa Fluor_488_-rabbit anti-mouse	5	ab196374	Abcam
IgG (H + L) highly cross-adsorbed secondary antibody	Alexa Fluor_555_-donkey anti-rabbit	5	A-31572	Thermo Fisher Scientific
IgG1	Rat	5	0116-01	Southern Biotech, Birmingham, AL, USA
IgG2a		5	0117-01	
IgG2b		5	0118-01	
IgG	Rabbit	0.5	0111-01	

**Table 2 biomedicines-12-01672-t002:** Recipe for AEC buffer.

Reagents	Quantity	Cat. No	Source of Buffer Component
3-amino-9-ethylcarbazole	1 tablet	#A6926	Sigma-Aldrich
dimethylformamide	7.5 mL	#1.03034	Merck Millipore
H_2_O_2_	150 μL	#1072090250	Merck Millipore
acetate buffer	15 mL, 1 mol/L, pH 5.5	UMCG, Pathology and Medical Biology department, in-house prepared	
deionized water	135 mL		

**Table 3 biomedicines-12-01672-t003:** Recipes for antigen retrieval buffers.

Reagents	Total Volume	Quantity Used	pH	Source of Buffer Component
1 mM EDTA	1 L demi water	Na_2_-EDTA, 372 mg	8.0	E1644, Lot#SLBF8074V, Sigma
10 mM Tris/1 mM EDTA		Tris Base, 1.21 g	9.0	648311, Lot#3244015, Millipore
		Na_2_-EDTA, 372 mg		E1644, Lot#SLBF8074V, Sigma
0.1 M Tris/HCl		Tris Base, 12 g	9.0	648311, Lot#3244015, Millipore

**Table 4 biomedicines-12-01672-t004:** Assay-on-demand primers used to determine mRNA levels of genes of interest. All primers were purchased from Thermo Fisher Scientific.

Table	Gene ID	Assay ID	Encoded Protein
*Sele*	20339	Mm00441278_m1	E-selectin
*Vcam1*	22329	Mm00449197_m1	Vascular cell adhesion protein 1
*Ptprc*	19264	Mm01293577_m1	Protein tyrosine phosphatase, receptor type, C (a.k.a. Cluster of differentiation CD45)
*Cd68*	12514	Mm03047343_m1	Cluster of Differentiation 68
*Gapdh*	14433	Mm99999915_g1	Glyceraldehyde 3-phosphate dehydrogenase

## Data Availability

The authors confirm that the data supporting the findings of this study are available within the article.

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
