# Peer review of "Heterogeneous Patterns of Endothelial NF-κB p65 and MAPK c-Jun Activation, Adhesion Molecule Expression, and Leukocyte Recruitment in Lung Microvasculature of Mice with Sepsis"

_biomedicines, 2024, doi:10.3390/biomedicines12081672_

Round 1
Reviewer 1 Report
Comments and Suggestions for Authors
In the manuscript entitled “Heterogeneous patterns of endothelial NF-κB p65 and MAPK c-Jun activation, adhesion molecule expression, and leukocyte recruitment in lung microvasculature of mice with sepsis”, the authors explored the nuclear localization of p65 and c-Jun in lung endothelial cells and expression of E-selectin, VCAM-1 and leukocyte subset recruitment in CLP-induced sepsis model. The results indicated that p65 and c-Jun were located in a subset of the lung endothelial cell nucleus in response to sepsis. E-selectin was mainly expressed in a subset of microvessels and VCAM-1 was expressed in a scattered pattern in alveolar tissue and microvessels. Neutrophil and T lymphocyte recruitment was observed at 4 and 7 hours after CLP surgery. Based on immunohistochemistry, neutrophil infiltration occurred in all pulmonary microvascular beds. In contrast, T lymphocytes were present in alveolar space and postcapillary venules. Overall, this is an interesting and foundational study, which could be beneficial for the field. The experiments were well-designed and conducted. The conclusions are well supported by the results and are solid. However, several concerns need to be addressed.
1. The study is largely based on immunohistochemistry. Therefore, it would be better to quantify the data to improve scientific rigor and reproducibility.
2. The authors concluded that “nuclear localization of p65 and c-Jun in EC and neutrophil recruitment were associated with induced E-selectin expression in the pulmonary microvessels in CLP-septic mice at the early stage of the disease”. It is overstated because no evidence supports this hypothesized association in the study. Thus, it would be better using “could be associated with”.
3. Please include the information on animal study protocol in the Method section.
4. No scale bar in Figure 1B.
Author Response
In the manuscript entitled “Heterogeneous patterns of endothelial NF-κB p65 and MAPK c-Jun activation, adhesion molecule expression, and leukocyte recruitment in lung microvasculature of mice with sepsis”, the authors explored the nuclear localization of p65 and c-Jun in lung endothelial cells and expression of E-selectin, VCAM-1 and leukocyte subset recruitment in CLP-induced sepsis model. The results indicated that p65 and c-Jun were located in a subset of the lung endothelial cell nucleus in response to sepsis. E-selectin was mainly expressed in a subset of microvessels and VCAM-1 was expressed in a scattered pattern in alveolar tissue and microvessels. Neutrophil and T lymphocyte recruitment was observed at 4 and 7 hours after CLP surgery. Based on immunohistochemistry, neutrophil infiltration occurred in all pulmonary microvascular beds. In contrast, T lymphocytes were present in alveolar space and postcapillary venules. Overall, this is an interesting and foundational study, which could be beneficial for the field. The experiments were well-designed and conducted. The conclusions are well supported by the results and are solid. However, several concerns need to be addressed.
Comments 1: The study is largely based on immunohistochemistry. Therefore, it would be better to quantify the data to improve scientific rigor and reproducibility.
Response 1: We fully agree with this reviewer that semi-quantitation is preferred over the descriptive approach we eventually had to take. We would like to share with you the path taken, to explain how we got to the conclusion that semi-quantitation of the proteins and cells under study in microvascular segments is not possible at present.
To quantify p65 and c-Jun positive nuclei, we consulted experts in the UMCG Microscopy and Imaging Center about the best way to quantify them in the scans of the sections of the mouse lungs. As per their advice, we used different software packages for this purpose, including Image J, TissueFAXS Imaging Software, and Aperio ImageScope, yet none of the programs could distinguish each unique positive nucleus. A major hurdle was that if 2 positive nuclei were located close to each other, the program would recognize these 2 as 1 nucleus. Colleagues at the Pathology Department confirmed the difficulty in assessing such staining results with software, irrespective of the extensive effort they have put into semi-automating this workflow for diagnostic purposes. Next, we tried to hand count positive p65 nuclei in at least 5 rectangles of 240 μm × 391 μm. However, our eyes could not identify each positive nucleus as extensive cytoplasmic staining would hide to some extent the nuclear staining. The results of two independent observers blinded to the mouse number-experimental condition key were not trustworthy. A similar path was taken to investigate options for quantitation of the leukocyte subsets, bringing us to the same conclusion that (semi)quantitation thereof is not (yet) possible. Advice from an immunologist to quantitate leukocyte influx by isolating the cells followed by antibody-based staining and flow cytometry-based analyses, could not be followed up on, as we do not have fresh material available.
For the adhesion molecules, expression was confined to dedicated microvasculature, which from our perspective did not need a (semi)quantitative measure to answer our research questions posed.
As we do agree with this reviewer that this is an important issue to share with the readers, we have added the following sentence to the Discussion section to address this issue:
In Iines 456-461 of the revised version: In addition, we attempted to quantify nuclear p65 and c-Jun proteins, as well as leukocyte subsets. As neither hand-counting nor software programs (Image J, TissueFAXS Imaging Software, Aperio ImageScope) could accurately recognize each positive nucleus respectively cell, and for leukocyte subset semi-quantitation by flow cytometry no fresh tissue was available, a descriptive method was taken to report these parameters.
Comments 2: The authors concluded that “nuclear localization of p65 and c-Jun in EC and neutrophil recruitment were associated with induced E-selectin expression in the pulmonary microvessels in CLP-septic mice at the early stage of the disease”. It is overstated because no evidence supports this hypothesized association in the study. Thus, it would be better using “could be associated with”.
Response 2: We agree with this reviewer that we do not have direct evidence to support a causal relationship between activation of signaling pathways, neutrophil recruitment, and E-selectin expression. Thus, we changed this description based on your suggestion in line 42 of the revised version.
Comments 3: Please include the information on animal study protocol in the Method section.
Response 3: Due to word limitations, we could not provide extensive information in the Methods section of the abstract. Yet, based on your comment, we re-wrote the Methods of the abstract and added information about the time period within which we studied the different parameters.
We have adapted the text as follows (lines 27~33 of the revised version):
Mice underwent cecal ligation and puncture (CLP) to induce polymicrobial sepsis and were sacrificed at different time points up to 72 hours after sepsis onset. Immunohistochemistry and reverse transcription-quantitative polymerase chain reaction (RT-qPCR) analyses were used to determine the kinetics of nuclear localization of p65 and c-Jun in EC, expression and location of adhesion molecules E-selectin and vascular cell adhesion molecule 1 (VCAM-1). Furthermore, extent and location of leukocyte recruitment were assessed based on Ly6G staining of neutrophils, cluster determinant (CD) 3 staining of T lymphocytes, and CD68 staining of macrophages.
Comments 4: No scale bar in Figure 1B.
Response 4: We apologize for the small-scale bar in the original figure, based on your comment we improved its readability.
Reviewer 2 Report
Comments and Suggestions for Authors
Review report June 28, 2024:
The submitted manuscript entitled” Heterogeneous patterns of endothelial NF-κB p65 and MAPK c-Jun activation, adhesion molecule expression, and leukocyte recruitment in lung microvasculature of mice with sepsis” is well designed article and has interesting data. However, some general issues were raised during the review process such as some grammatical mistakes need to be checked all over the manuscript. The percentage of recent references (2020-present) is low, less than 20%.
1- Abstract: All abbreviated terms must be fully mentioned when used at first time.
2- Abstract, methods: How were lymphocytes and neutrophils identified?
3- Line 61-64 (L61-640): It is not clear what exact signals were involved? Please add brief description.
4- L65 – 69: General information, it is recommended that you be more specific and introduce relevant information to you work.
5- L102: How authors can ensure that the squeezed cecal material was 3mm? Also, why it is only 3mm?
6- Section 2.1: Why was isoflurane used then followed by s.c Buprenor, before surgery? Please justify this in the manuscript.
7- Post-surgery care should be fully demonstrated.
8- Even the samples were taken from previous work, ethical statement should be declared again in this section.
9- L102 - 106: The statements of animal grouping and endpoint of the experiment are confusing. Control, Sham and CLP groups were euthanized at 4, 7 and 24h respectively???? then 72 h post surgery?? Please restate this carefully.
10- Based on your previous work, antibiotics were used, don’t you think this could delay/avoid/affect the severity of sepsis?
11- Some results show lung study at 4, 7, 24 and some additional time at 72 is taken too.
12- Sham group should be considered and included in all results.
13- L154: the examining of sections was performed by eyes?
14- Dr. Derek Brown should be acknowledged too in acknowledgment section.
15- L162-164: you may justify why you couldn’t differentiate between microvessels. Is this standard phenomenon? Please reference it.
16- L184: for what “respectively” is used here? This mistake is repeated four times in different places.
17- L186: What do you mean with data not shown? For example, figure 2b red and green are presented. Please clarify this.
18- L189: Why total RNA was extracted from cryopreserved samples not from freshly collected lungs?
19- L190: please separate ref 11 from previously.
20- L199: if you calculated delta-delta Ct, then you calculated normalized gene expression, not relative. Which one is correct?
21- Table4: the assay ID should be double checked. However, the primer IDs should be following standard identifications. For example, NCBI accession number should be listed. Also, the primer sequences should be listed too.
22- If I’m not mistaken, results of CD31, Pecam1, were not presented nor discussed.
23- L519 and L540: how much do authors sure that the LPS is the only stimulant in inducing sepsis in this model?
Comments on the Quality of English LanguageSome mistakes of using respectively was mentioned in the review report.
Author Response
Comments 1: The submitted manuscript entitled” Heterogeneous patterns of endothelial NF-κB p65 and MAPK c-Jun activation, adhesion molecule expression, and leukocyte recruitment in lung microvasculature of mice with sepsis” is well designed article and has interesting data. However, some general issues were raised during the review process such as some grammatical mistakes need to be checked all over the manuscript. The percentage of recent references (2020-present) is low, less than 20%.
Response 1: Regarding the above comment on that we used <20% of recent publications used:
Endothelial cell activation and interactions with leukocytes have been a major subject of investigation since the mid-1990s into the 2000s, revealing major molecular concepts and insights. EC heterogeneity, between organs, and within organs between microvascular segments, has, however, been seen as a major hurdle for the progress of the field, especially since this heterogeneity cannot be maintained when EC from an organ are put into culture (Cleuren et al, Proc Natl Acad Sci U S A., 2019, PMID: 31712416). We feel we stand on the shoulders of those who were brave and visionary to start working on the molecular control of EC responses to all kinds of stimuli and hence work with their published work as a foundation for our molecular knowledge. In the new area of scRNA sequencing new molecular insights are being generated at an extremely fast pace, and application thereof to unveil molecular paradigms of in vivo EC responses in e.g., sepsis onset and development, is awaited.
Comments 2: Abstract: All abbreviated terms must be fully mentioned when used at first time.
Response 2: Thank you for your suggestion. Throughout the abstract, we have added words in full for abbreviations used.
In lines 23~25: […] endothelial nuclear factor kappa-light-chain-enhancer of activated B cells (NF-κB) p65 and mitogen-activated protein kinase (MAPK) c-Jun intracellular signal transduction pathways […]
Comments 3: Abstract, methods: How were lymphocytes and neutrophils identified?
Response 3: In this study, we used antibodies to recognize surface antigen CD3 (T lymphocytes) and Ly6G (neutrophils) and CD68 (macrophages). This information has been added to the Methods section of the abstract (revised version shown in full below):
Mice underwent cecal ligation and puncture (CLP) to induce polymicrobial sepsis and were sacrificed at different time points up to 72 hours after sepsis onset. Immunohistochemistry and reverse transcription-quantitative polymerase chain reaction (RT-qPCR) analyses were used to determine the kinetics of nuclear localization of p65 and c-Jun in EC, expression and location of adhesion molecules E-selectin and vascular cell adhesion molecule 1 (VCAM-1). Furthermore, extent and location of leukocyte recruitment were assessed based on Ly6G staining of neutrophils, cluster determinant (CD) 3 staining of T lymphocytes, and CD68 staining of macrophages.
Comments 4: Line 61-64 (L61-64): It is not clear what exact signals were involved? Please add brief description
Response 4: We refer here to the NF-κB and MAPK pathways as described in the references used. A brief description about these 2 pathways has been added to lines 64-66 in the revised version, and some additional changes regarding this issue were also made in line 75.
Comments 5: L65-69: General information, it is recommended that you be more specific and introduce relevant information to your work.
Response 5: As per this reviewer’s request, we have rewritten these introductory sentences toward a focus on what happens with EC once LPS binds to toll-like receptor 4:
In lines 70-74 of the revised version: In EC, LPS and sepsis-related systemically released pro-inflammatory cytokines such as tumor necrosis factor-alpha (TNF-α), activate a complex pattern of kinases[9] which eventually results in the accumulation of transcription factors in the nucleus. Here they initiate the transcription of DNA sequences that code for, among others, adhesion molecules[10].
Comments 6: L102: How authors can ensure that the squeezed cecal material was 3mm? Also, why it is only 3mm?
Response 6: We used one puncture to ensure we had a good view on how much stool we squeezed out of the cecum. Three mm was used to standardize the amount of starting material leading to infection. We did not measure exactly 3 mm but trained the animal experimenters to try and stick to this size as accurately as possible. To clarify this approximation approach, we added the word ‘approximately’ in line 104 of the revised version.
Comments 7: Section 2.1: Why was isoflurane used then followed by s.c Buprenor, before surgery? Please justify this in the manuscript.
Response 7: Isoflurane is an anesthetic to make the mice be asleep before the surgery starts, while buprenorphine is a pain killer. The buprenorphine is given at the start of surgery, so the mice will have minimal pain, with the pain killing starting when they regain consciousness immediately after the operation.
We have added this description in the adapted version in lines 100 and 102 of the revised version:
[…] after anesthetization by isoflurane inhalation, mice were subcutaneously injected with Buprenorphine (0.1 mg/kg, Buprecare, ASTFarma, Oudewater, The Netherlands) to reduce the pain of the mice post-surgery.
Comments 8: Post-surgery care should be fully demonstrated.
Response 8: As per this reviewer’s request, we have added more details after post-surgery care in lines 108-113 of the revised version.
After surgery, all mice were immediately s.c. injected with 0.5 mL saline on each abdominal side and then moved to individual 37 â—¦C heated cages for 30 min to recover from surgery, followed by being placed in normal housing conditions. Sham and CLP mice were given access to liquid food. Animals in the groups surviving beyond the 7 hours, were given antibiotics (Imipinem/Cilastatin, 25 mg/kg, Fresenius Kabi, Bad Homburg vor der Höhe, Germany) and Buprenorphine (0.1 mg/kg) at 10 hours after surgery by s.c. injection.
Comments 9: Even the samples were taken from previous work, ethical statement should be declared again in this section.
Response 9: Thank you for this comment. Based on the requirements of Biomedicines, this information should go to the section of the Institutional Review Board Statement. Please see it in lines 544-546 of the revised version.
Comments 10: L102 - 106: The statements of animal grouping and endpoint of the experiment are confusing. Control, Sham and CLP groups were euthanized at 4, 7 and 24h respectively???? then 72 h post-surgery?? Please restate this carefully.
Response 10: We apologize for the confusion. We have rewritten the statement about the groups as follows in line 115 of the revised manuscript: ‘Mice in sham and CLP groups were sacrificed at 4, 7, 24, or 72 hours after surgery’.
Comments 11: Based on your previous work, antibiotics were used, don’t you think this could delay/avoid/affect the severity of sepsis?
Response 11: Thank you for your question. The addition of antibiotics is unlikely to influence the model. They were given 10 hours after surgery so there was enough time for the mice to develop sepsis, as witnessed by the fact that already in the first hours after surgery the CLP mice showed obvious signs of illness. Besides the fact that in our country an animal experimentation set-up like this requires antibiotics treatment, we also wanted to simulate the clinical situation in which a septic patient is often started on antibiotics treatment a number of hours after the onset of disease.
Comments 12: Some results show lung study at 4, 7, 24 and some additional time at 72 is taken too.
Response 12: This is correct. The reason for this was that the results of the p65 and c-Jun staining showed that most discernible differences compared to sham groups occurred at 4, 7, and 24 hours after CLP-sepsis initiation. 72 hours after CLP-sepsis started, nuclear staining of p65 and c-Jun was comparable to that in control and sham mice. Thus, we put the data of nuclear p65 and c-Jun staining of the lung samples of the mice sacrificed at 72 hours after CLP surgery in supplementary figure 1.
Comments 13: Sham group should be considered and included in all results.
Response 13: Thank you for your suggestion. As the extent of the nuclear p65 and c-Jun staining at all time points in sham mice was comparable to that in control mice, we only showed the result of control mice in main figures and put all sham mice data in supplementary figure 1.
In the revised version, we amended the brief description of sham groups in lines 255-257: In the first 24 hours after sham operation, the extent of p65-positive nuclei was comparable to that in the control group (Figure S1A).
In lines 285-286: Following CLP-induced sepsis, more nuclei became c-Jun positive, in alveolar tissue and postcapillary venules this increase was only observed at 4 hours post-CLP, while scarce positive nuclei were present in the sham group (Figure 3A&Figure S1B).
In lines 289-290: At 72 hours after CLP-induced sepsis started, nuclear c-Jun in the lung returned to basal levels comparable to those in the control and sham groups (Figure S1B).
In line 319: In all sham groups, E-selectin expression was absent at all time points (Figure 4B).
In lines 383-384: In all sham groups, sparse neutrophils were present in alveolar tissue and postcapillary venules (Figure 7).
In line 395: In all sham groups, sparse T lymphocytes were present in alveolar tissue (Figure 8).
Comments 14: L154: the examining of sections was performed by eyes?
Response 14: If we understand correctly, you ask us how we evaluated the scans of lung sections.
As we described in section 2.3 ‘Scoring of Immunohistochemical Staining’, we visually inspected the scanned sections by eye. In section 2.3, we described to focus on the microvasculature and explained a number of choices made. We have added the following to that section:
In lines 174~182 of the revised version.
To analyze the extent of p65 and c-Jun staining, at least 3~5 arterioles or postcapillary venules were evaluated in each section. To determine staining in alveolar tissue, encompassing alveolar capillaries as well as other cells, at least 10 rectangle areas (240 μm × 390 μm) were randomly selected and evaluated. The scoring values given were absent (no nuclei stained), intermediate (minority of nuclei stained), or extensive (majority of nuclei stained). E-selectin and VCAM-1, as well as Ly6G, CD68, and CD3 staining were also qualitatively scored by visually assessing the scans, with values given being absent, low, or present (cell adhesion molecules)/high infiltration (leukocyte subset). Results per group were summarized.
Ideally, we would have used software programs (Image J, TissueFAXS Imaging Software, Aperio ImageScope) to identify positive staining, however, these software programs were not able to discriminate 2 adjacent positive nuclei or cells. Instead, they counted this as 1 nucleus or cell, thus we concluded this was not an accurate method to use. In hand counting, our eyes could not identify each positive nucleus as extensive cytoplasmic staining would hide to some extent the nuclear staining. The results of two independent observers blinded to the mouse experimental conditions were therefore in the end also not trustworthy.
Comments 15: Dr. Derek Brown should be acknowledged too in the acknowledgment section.
Response 15: Thanks for your suggestion. We have added this information to the acknowledgment section in lines 552-554 of the revised version: We also thank Dr. Derek Brown (UCB S.A., Brussels, Belgium) for his kind gift of the E-selectin antibody producing hybridoma clone MES-1.
Comments 16. L162-164: you may justify why you couldn’t differentiate between microvessels. Is this standard phenomenon? Please reference it.
Response 16: Pulmonary microvascular arterioles were discriminated from postcapillary venules based on visible elastic lamina(s), collagen, and smooth muscle cell (SMC) layers in the former, whereas the latter was devoid of a collagen layer and contained a thinner SMC layer than arterioles. The above features of arterioles and postcapillary venules are easily distinguished in FFPE samples, yet difficult to discriminate in cryosection. Since staining of EC cell adhesion molecules could only be done using cryosections, we called them ‘microvessels’ when referring to these samples. An experienced lung pathologist in UMCG confirmed the difficulty in distinguishing pulmonary arterioles and venules in frozen tissue. While we would be willing to introduce a reference to this issue, we have not come across such a reference, unfortunately, hence we decided to describe this issue in a bit more detail in Section 2.3 in lines 164~173 of the revised version.
Comments 17: L184: for what “respectively” is used here? This mistake is repeated four times in different places.
Response 17: We apologize for making this mistake. We have deleted this word in the revised manuscript in lines 115, 202, 320, and 351.
Comments 18: L186: What do you mean with data not shown? For example, figure 2b red and green are presented. Please clarify this.
Response 18: In immunofluorescence double staining, we used Ab based green (ERG) and red (c-Jun or p65), and nuclear blue (DAPI). Overlay of green, red, and blue would give white as a result, based on which we would conclude that green and red colocalized in the nucleus.
In the last part of section 2.4, we describe control analyses (lines 200-204 of revised version) to confirm signal specificity, i.e., that the signal we obtained is indeed from the one, or the other Ab.
In this setup, we added one pair of Ab-based fluorescence signals (1st and/or 2nd Ab based) on the tissue section yet determined the signal in all three channels. Would one of the Ab (pairs) show non-specific binding in the tissue section or have its fluorescence label leak into the channel of the other Ab (pair) based signal, we would wrongly conclude that colocalization occurred. We showed with these (one label missing) control analyses that non-specific binding or signal leakage did not occur (data not shown).
In Figure 2B, the data is created by (p65+Alexa-555)+ERG-488 plus DAPI. To clarify the signal in each channel for the reader, we decided to separate the signals in all channels and display red, green, blue, and merged colors.
Comments 19: L189: Why total RNA was extracted from cryopreserved samples not from freshly collected lungs?
Response 19: In our protocol for collecting the organs including lungs, once we sacrificed the mice, we immediately took all organs out and snap-freeze them on liquid nitrogen and stored them at -80 °C. This workflow and storage conditions support the total RNA integrity of mouse lungs (Auer et al, Biotech Histochem., 2014, PMID: 24799092), which we confirmed by agarose gel-based RNA integrity analysis (line 207 of revised version)
Comments 20: L190: please separate ref 11 from previously.
Response 20: We apologize for this mistake. We have changed it in line 208.
Comments 21: L199: if you calculated delta-delta Ct, then you calculated normalized gene expression, not relative. Which one is correct?
Response 21: We apologize for the confusion. In lines 213~217 in the revised manuscript, we have written the following to clarify: Once the average quantification cycle (Cq) values were obtained via QuantStudio Real-Time PCR software, expressed genes were normalized to the reference gene Gapdh, resulting in the ∆Cq value. The relative mRNA level was calculated using 2-∆Cq.
Comments 22: Table 4: the assay ID should be double-checked. However, the primer IDs should be following standard identifications. For example, the NCBI accession number should be listed. Also, the primer sequences should be listed too.
Response 22: Thank you for your suggestion. The assay IDs have been double-checked and approved. NCBI accession numbers for each gene have been added to the new Table 4. As the primer sequences are not released to customers by the company that designs TaqMan primers, we also do not know the concrete primer sequences.
Comments 23: If I’m not mistaken, results of CD31, Pecam1, were not presented nor discussed.
Response 23: Indeed, we did not show CD31 staining results nor Pecam1 mRNA data in our main figures. We have deleted the information on the Ab and primers from the manuscript.
For your information: to identify pulmonary microvascular beds, we stained sequential cryo-sections by immunohistochemistry with anti-CD31 to identify the location of microvessels (see also R2Q5), but did not further elaborate on this in the manuscript.
Also, Pecam1 mRNA data are only used by our research team as an experimental control to check whether samples have similar levels of endothelial cell content.
Comments 24: L519 and L540: how much do authors sure that the LPS is the only stimulant in inducing sepsis in this model?
Response 24: In line 490 of the revised version, we refer to in vitro studies in which EC were challenged with LPS, in line 511 of the revised version, the context is an endotoxemia model, so an in vivo experiment. While in vitro EC hardly make any pro-inflammatory cytokines themselves - so, the LPS stimulus is rather clean – LPS administration in vivo leads to a cytokine storm, as nicely described in the paper by Seemann et al (Seemann et al, J Biomed Sci., 2017, PMID: 28836970).
In mice with CLP-induced sepsis, as used in our study, bacteria induce the production of pro-inflammatory cytokines such as TNF-α and IL-6, partly by shedding LPS in the systemic circulation (Villa et al, Clin Diagn Lab Immunol., 1995, PMID: 8548533). The notion in the field is that the pathophysiological manifestation of sepsis in CLP mice resembles human sepsis (Seemann et al, J Biomed Sci., 2017, PMID: 28836970).
Comments 25: Comments on the Quality of English Language
Some mistakes of using respectively were mentioned in the review report.
Response 25: we apologize for making this mistake. We have changed or deleted this word in the revised manuscript version.
Round 2
Reviewer 2 Report
Comments and Suggestions for Authors
All concerns raised in previous version are sufficiently addressed in this version. Thanks